# A novel expert-annotated single-cell dataset for thyroid cancer diagnosis with deep learning benchmarks

Nguyen Quang Huy[1]☯, Thanh-Ha Do[2]☯*, Nguyen Van De[3],
Hoang Kim Giap[1], Vu Huyen Tram[1]

**1** Faculty of Mathematics Mechanics and Informatics, VNU University of Science, Hanoi, Vietnam,,
**2** Faculty of Artificial Intelligence, Posts and Telecommunications Institute of Technology, Hanoi, Vietnam,,
**3** Military Central Hospital, Hanoi, Vietnam

☯ These authors contributed equally to this work.
* dothanhhha@ptit.edu.vn

## Abstract

This paper introduces a novel, expert-annotated single-cell image dataset for thyroid cancer diagnosis, comprising 3,419 individual cell images extracted from high-resolution histopathological slides and annotated with nine clinically significant nuclear features. The dataset, collected and annotated in collaboration with pathologists at the 108 Military Central Hospital (Vietnam), presents a significant resource for advancing research in automated cytological analysis. We establish a series of robust deep-learning baseline pipelines for multi-label classification on this dataset. These baselines incorporate ConvNeXt, Vision Transformers (ViT), and ResNet backbones, along with techniques to address class imbalance, including conditional CutMix, weighted sampling, and SPA loss with Label Pairwise Regularization (LPR). Experiments evaluate the good performance of the proposed pipelines, demonstrating the challenges over the dataset's characteristics and providing a benchmark for future studies in interpretable and reliable AI-based cytological diagnosis. The results highlight the importance of effective model architectures and data-centric strategies for accurate multi-label classification of single-cell images.

## Author summary

Deep learning approaches to cancer diagnosis over cell images are often trained to directly classify cells as either 'cancerous' or 'non-cancerous.' Although these models can achieve high accuracy, they typically do not provide interpretable reasoning behind each prediction, which may limit their clinical trustworthiness. In this study, we propose an alternative strategy that emphasizes transparency and interpretability. We introduce a new dataset consisting of cell images annotated by medical experts with nine distinct morphological attributes associated with

**Data availability statement:** The complete dataset is publicly available on Zenodo at 10.5281/zenodo.17226934 To ensure full reproducibility, we have also released the entire codebase, including baseline implementations, the proposed hybrid pipeline, pretrained model weights, and comprehensive documentation, at https://github.com/nqHuynq/hybrid-cell-classification Pretrained models are additionally hosted on Hugging Face (as referenced in the GitHub repository) to facilitate straightforward reuse and extension in future research.

**Funding:** The author(s) received no specific funding for this work.

**Competing interests:** The authors have declared that no competing interests exist.

malignancy. Rather than predicting binary cancer labels, our AI model is trained to understand the presence or absence of these attributes, allowing for a more granular and interpretable analysis. This feature-centric approach enhances diagnostic clarity and may assist clinicians by offering objective, biologically grounded insights into cellular abnormalities, potentially facilitating human-AI collaboration in diagnosis.

## 1 Introduction

Accurate cancer diagnosis begins at the cellular level [1], yet interpreting cell morphology [2] remains a complex and subjective task, even for experts. Cytological analysis [2,3], which involves the microscopic examination of cellular features, plays a vital role in detecting malignant or precancerous conditions. However, this process is inherently reliant on expert interpretation, leading to diagnostic variability and limited scalability across healthcare settings [4].

Recent strides in computer vision are powering the automation of cytological assessment [5], leading to diagnostics that are inherently more consistent, rapid, and scalable. Nevertheless, automating this task remains particularly challenging due to the multi-label nature of cytological classification [6], where a single cell may exhibit several abnormal traits simultaneously, as well as class imbalance, label correlation, and visual heterogeneity arising from diverse staining protocols and imaging conditions.

To support the development and evaluation of automated solutions, we introduce a novel single-cell cytology dataset for multi-label classification, curated explicitly for thyroid cancer analysis. The dataset was constructed through a three-stage pipeline [7] involving (1) expert annotation of regions of interest on cytological slides, (2) instance segmentation and detailed labeling of individual cells with nine clinically significant nuclear features, and (3) standardized extraction of single-cell images centered on a 224×224 white background. The dataset comprises 3,419 cell images derived from 250 detailed thyroid cytology images.

However, this dataset presents several inherent challenges common in medical image analysis. First, the distribution of samples across the nine nuclear features is highly imbalanced [8], with some features rarely occurring while others are prevalent. This class imbalance can bias model training and hinder the detection of rare but clinically important features accurately. Second, the multi-label classification task inherently involves complex label dependencies and co-occurrences that require models to identify individual features and understand the relationships between morphological characteristics. Moreover, two critical factors constrain the AI model's effectiveness: the dataset is limited to encompass the task's complexity, and similarities between classes frequently lead to misclassification.

In this study, we present this dataset to the research community and establish a series of standardized deep learning baseline pipelines for multi-label classification on this dataset. These pipelines share a unified experimental setup to ensure

fair comparison and incorporate several deep learning architectures, including ConvNeXt, Vision Transformers (ViT), and ResNet-50 [9]. Alongside these backbones, we explore various data augmentation strategies, loss functions, and sampling techniques tailored to address the challenges posed by class imbalance and multi-label dependencies [10]. We aim to facilitate future research and development in automated cytological analysis by comprehensively evaluating these configurations through an ablation study.

### 1.1 Key contributions

The key contributions of this paper are as follows:

1. A novel, expert-annotated single-cell image dataset for thyroid cancer diagnosis, processed through a three-stage pipeline comprising 3,419 individual cell images extracted from 250 high-resolution histopathological slides with detailed annotations for nine clinically significant nuclear features.
2. Establishes and compares the performance of some deep learning pipelines for multi-label classification on this dataset, specifically for handling class imbalance.
3. Detailed analysis of the performance of these baselines, providing insights into the challenges and opportunities for automated cytological analysis.

## 2 Literature reviews

### 2.1 Multi-label classification in biomedical imaging

Multi-label classification has become increasingly important in biomedical image analysis, where a single image may be associated with multiple biological or pathological features [14]. In contrast to single-label classification, multi-label tasks in medical domains demand models capable of capturing complex semantic relationships among labels [16], which often co-occur or exhibit subtle dependencies. This is especially critical in cytology, where the morphology of a single cell may simultaneously express multiple traits relevant to cancer diagnosis. Deep convolutional neural networks (CNNs) [11] have been widely adopted to address this challenge by using sigmoid activations and training with binary cross-entropy (BCE) loss [13] to produce independent probabilities for each label [12,15]. However, such baseline methods often struggle to generalize in realistic clinical datasets, where label distribution is heavily skewed, and labels are not conditionally independent.

### 2.2 Class imbalance in cytology datasets

One of the most persistent challenges in multi-label learning—particularly in cytology and histopathology—is the imbalance in label distribution [8]. Some cell traits or pathologies are inherently rare, while others dominate the dataset due to their prevalence in clinical populations or acquisition bias [18]. This imbalance makes models biased toward the majority labels, resulting in high overall accuracy but poor recall on rare classes [8,17]. Furthermore, the imbalance becomes more nuanced in multi-label settings because rare labels often co-occur with common ones, complicating the decision boundary. The metrics like accuracy or average precision may fail to capture this discrepancy. Thus, designing and evaluating models need to be done more carefully.

### 2.3 Approaches to handling imbalance: Loss functions, sampling, and augmentation

Several strategies have been developed to mitigate the adverse effects of class imbalance in multi-label classification. Loss functions such as focal loss [20] re-weight easy and hard samples dynamically, emphasizing rare or misclassified labels more. Re-weighting schemes using inverse class frequency or an effective number of samples have also been proposed [21]. Sampling strategies, such as weighted random sampling or oversampling of rare-label instances, help balance label presence during training.

Data augmentation plays a complementary role in improving model generalization and rare-label learning. While traditional augmentations (e.g., rotations, flips, scaling) increase input diversity, more advanced techniques like CutMix [22] and MixUp [23] have shown promise in generating synthetic training examples that preserve label structure. CutMix is particularly useful in biomedical contexts as it respects spatial semantics [19] while promoting exposure to underrepresented patterns. Nevertheless, these methods require careful integration with loss functions and sampling schemes to avoid overfitting common patterns or diluting rare label signals.

## 2.4 Modern backbones for multi-label classification

In recent years, the backbone architecture has played a central role in determining the performance of multi-label classification models, especially in high-dimensional and complex domains such as medical imaging. Among the most competitive architectures are ResNet [24], ConvNeXt [9], and vision transformers [25], each bringing distinct inductive biases and representational capacities. ResNet remains a strong baseline due to its simplicity and effectiveness in feature extraction. ConvNeXt, a modernized CNN architecture, builds upon ResNet's principles while incorporating design choices inspired by transformers, such as larger kernel sizes, GELU activations [26], and layer normalization, achieving state-of-the-art results on several image classification benchmarks. On the other hand, vision transformers excel at capturing long-range dependencies and modeling label relationships through self-attention mechanisms [27], which can be particularly useful in multi-label tasks with strong inter-label correlation.

However, no single architecture has demonstrated consistent superiority across all biomedical datasets. Performance tends to be dataset-specific, influenced by data resolution, label sparsity, and inter-label dependencies. For instance, ViTs may underperform on small or data-limited medical datasets due to their weaker inductive bias and need for larger-scale pretraining [28,29]. Conversely, ConvNeXt often balances efficiency and accuracy, especially when coupled with task-specific adaptations like tailored loss functions and augmentation strategies [9,28]. As a result, the choice of backbone in multi-label biomedical classification remains an empirical question, motivating hybrid approaches and systematic evaluations across multiple dimensions of the learning pipeline.

## 2.5 Model calibration and inter-label dependency

In multi-label learning, especially with imbalanced data, raw model outputs (i.e., sigmoid probabilities) are often poorly calibrated [30], leading to unreliable predictions. Threshold tuning [32] is commonly used to convert these probabilities into binary decisions per label, but setting a single global threshold may not suffice due to label-specific variance. Recent research emphasizes the need for more principled approaches to calibration, including per-label thresholding or temperature scaling [31].

Beyond thresholding, newer methods explicitly model inter-label relationships. For instance, SPA loss (Self-paced Asymmetric Loss) [33] dynamically adjusts the learning pace for each label based on difficulty. In contrast, Label Pairwise Regularization (LPR) [33,34] encourages the model to respect known or learned label co-occurrence structures. These techniques enhance calibration and improve performance in settings where label correlation is strong, as in cancer cell characterization.

## 2.6 Summary and research gap

Despite these advances, existing studies often treat imbalance mitigation and calibration and label correlation modeling as isolated problems. Few works systematically integrate them into a unified pipeline tailored for highly imbalanced, multi-label biomedical datasets with high inter-label dependency. Moreover, most benchmarks are conducted on large-scale public datasets that may not reflect the complexity of real-world clinical data, such as cytology images annotated with rare, subtle cell traits.

This study proposes a hybrid deep learning pipeline designed to address these limitations by combining data-driven imbalance handling (e.g., CutMix, weighted sampling), structure-aware loss functions (SPA + LPR), and adaptive threshold tuning. Grounded in a real-world cytology dataset collected and annotated in collaboration with clinical experts, our approach aims to advance the state-of-the-art in robust, interpretable multi-label classification for cancer diagnosis.

## 3 Materials and methods

### 3.1 Ethics statement

The study protocol was approved by the Scientific Council of Central Military Hospital 108, which serves as the institutional review board for research approval (Decision No. 5262/QĐ-BV). The requirement for informed consent was waived as this research was a retrospective analysis of fully anonymized data.

### 3.2 Dataset description

The following Fig 1 describes the data preparation pipeline, divided into two main stages: (A) Segmentation Stage by collaborating researchers and (B) Cropping Phases by our team.

### 3.3 Segmentation stage

Phase A identified regions of interest and extracted individual cell coordinates from the original thyroid cell images. While simple classical image analysis techniques like thresholding, watershed segmentation, and clustering are frequently employed, they struggle with noisy images, high-intensity variations, and clustered or irregularly shaped cells. Although other models, like active contour models and graph-based segmentation, can mitigate some of these issues, the high computational cost, especially for images with dense cell nuclei, is a significant drawback of these approaches. Consequently, deep-learning models have gained considerable traction in medical image analysis. The two dominant deep learning approaches [39,40] are (1) using models such as U-Net with post-processing for segmenting individual cells and (2) leveraging specialized object detection-based segmentation models like Mask R-CNN.

Specific applications of deep learning in pathological image analysis include Sharma et al.'s [41] system for classifying stomach cancer from whole-slide images and Korbar et al.'s [42] method for colorectal polyps, demonstrating its widespread adoption by research organizations [43,45]. Elman Neural Networks (ENNs) [44] have also been used to construct computer-aided thyroid cytology diagnosis methods. Moreover, advanced tools such as Nuclei AIzer, featuring a

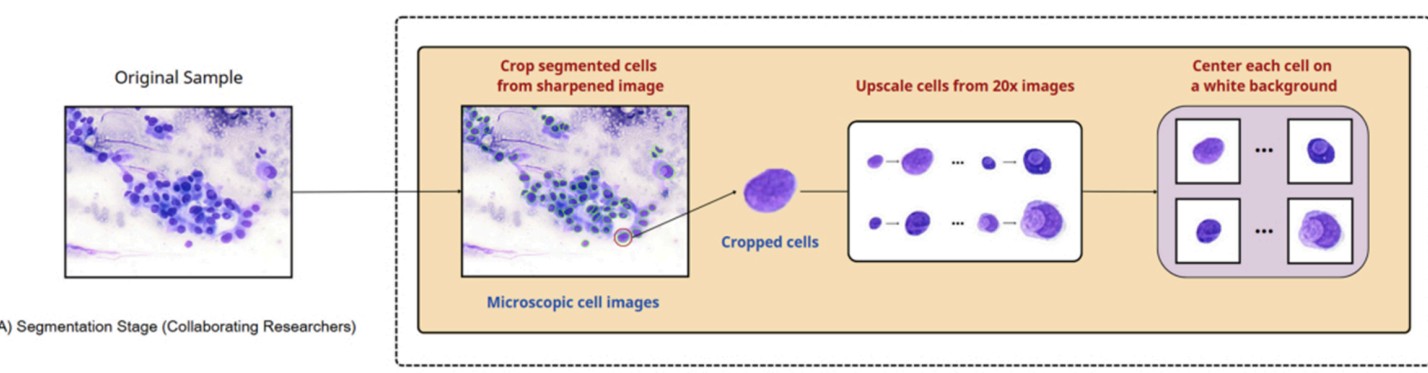

**Fig 1**. Data preparation process.

Mask R-CNN backbone improved by style transfer and U-Net for mask refinement, highlight the feasibility of fully automated, generalizable pipelines.

This research explores each cell's characteristics to provide more evidence to support the final decision. Thus, instance segmentation is required, and our experiments highlighted Mask R-CNNs for instance segmentation. We systematically evaluated models and improved segmentation accuracy by applying data post-processing and enhancement techniques to real-world thyroid cytology images.

Each of the segmented cell images goes through two steps: Cropping individual cells from sharpened microscopic images to isolate them, Upscaling the cropped cells from 20× images, and centering each cell on a 224×224 white background to create standardized inputs for model training.

### 3.4 Dataset overview

The final dataset comprises 3,419 single-cell images, each annotated with nine nuclear morphological features. Representative examples of these features are illustrated in Fig 2.

Two cytopathologists independently annotated a single-cell image following the features in Fig 2. When disagreements arose, a third senior pathologist adjudicated, and the final decision was established through group consensus. Although formal inter-observer agreement metrics were not computed due to resource constraints, this structured cross-validation

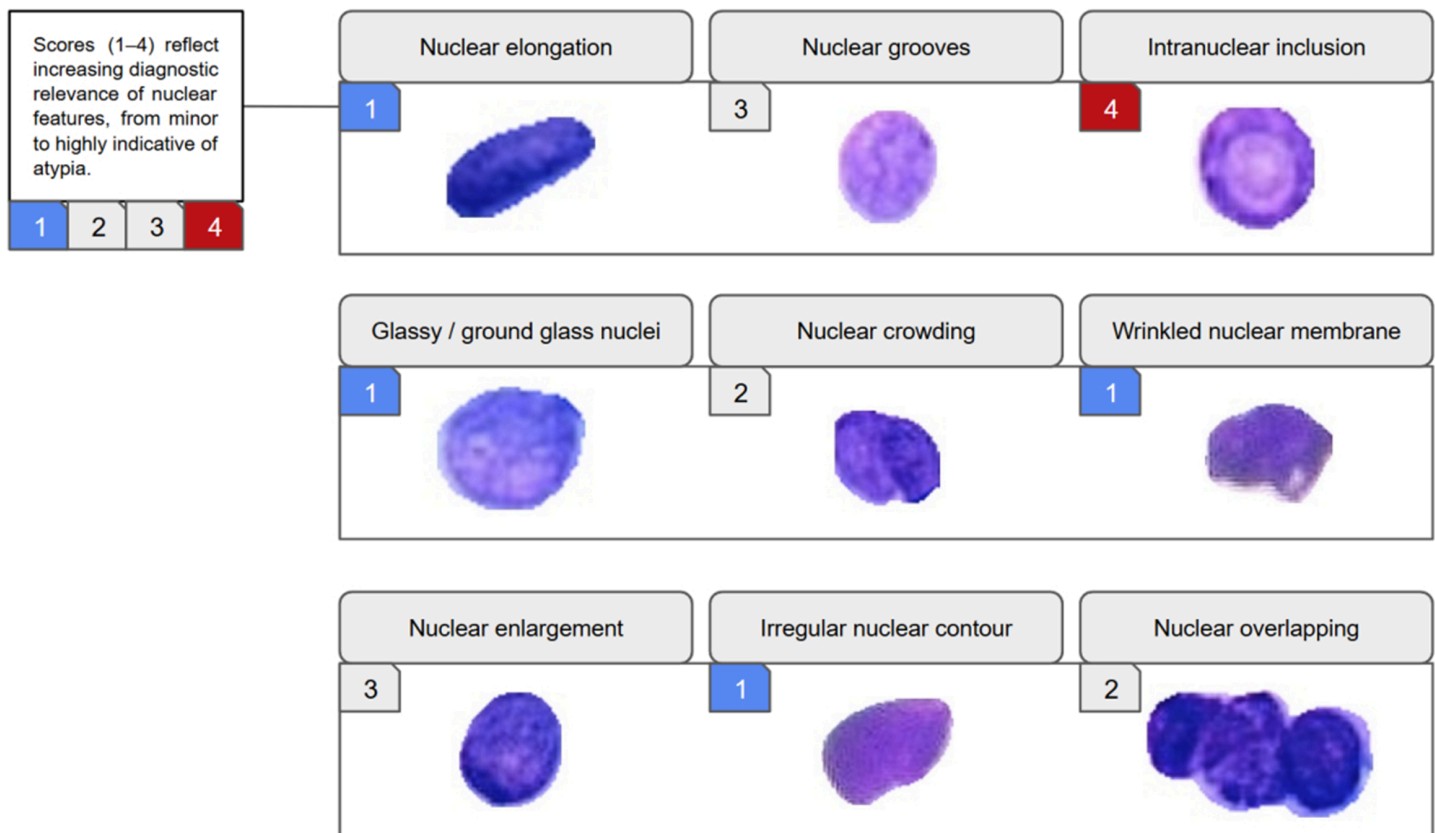

**Fig 2. List of nuclear morphological features annotated in the dataset.** Note: Each feature corresponds to a distinct nuclear morphology pattern associated with cellular atypia, commonly observed in cytopathological assessments. Medical experts annotated these features to support the multi-label classification task.

and adjudication protocol, supervised by the cytopathologist with more than 20 years of experience, maintained a high degree of consistency. This multi-expert process minimized individual bias and anchored the nine nuclear features in robust clinical consensus rather than subjective judgment.

To better understand the class imbalance inherent in the dataset, we analyze the frequency of each morphological feature across all annotated single-cell images. As shown in Table 1, the occurrence of features is unevenly distributed, with specific characteristics such as nuclear elongation and nuclear grooves being highly prevalent, while others, like intranuclear inclusion or irregular nuclear contour, are comparatively rare. This imbalance challenges supervised learning, especially in accurately identifying rare but clinically significant features.

### 3.5 Augmentation for rare class exposure

In multi-label pathology datasets, certain nuclear morphological traits—such as nucleoli enlargement or irregular contours—appear in less than 5% of the total samples. These underrepresented traits are referred to as rare classes, and their limited presence can hinder the model's ability to learn meaningful features for accurate recognition.

To address this imbalance, we employed CutMix augmentation [22]. This technique combines two training images by replacing a random region of one with a patch from another and interpolates their labels accordingly. For samples containing rare labels, CutMix was selectively applied to generate mixed examples where features of rare traits are presented in new contexts. This increased exposure helps the model learn more discriminative patterns associated with these underrepresented characteristics. The effectiveness of this strategy in improving rare class recognition is further evaluated in the Results section.

### 3.6 Model architecture and training configuration

This study employed the ConvNeXt [9] architecture as the backbone model. This study employed the ConvNeXt [9] architecture as the backbone model. We chose to develop ConvNeXt based on evaluation and compare it to another base pipeline such as ResNet50, Swin Transformer, EfficientNet-B3, ViT, DeiT, and Vit-B model (see Section Results).

ConvNeXt is a modern convolutional neural network (CNN) architecture that leverages the benefits of transformer-based architectures while maintaining the computational efficiency of traditional CNNs. It has achieved state-of-the-art performance on various image classification tasks, making it suitable for handling complex biomedical image classification problems, such as this study's multi-label cancer cell image classification task.

The ConvNeXt model was pre-trained on the *ImageNet1K_V1* dataset, providing a strong feature extraction initialization. The pre-trained weights were fine-tuned on our specific dataset to adapt the model to the unique characteristics of biomedical images, where the relationships between classes can be highly nuanced and interdependent. For the

**Table 1**. Label frequency distribution across the 9 morphological attributes.

| Feature | Label 0 | Label 0 (%) | Label 1 | Label 1 (%) |
|---|---|---|---|---|
| Nuclear elongation | 1046 | 29.96 | 2445 | 70.04 |
| Nuclear grooves | 1315 | 37.67 | 2176 | 62.33 |
| Intranuclear inclusion | 3279 | 93.93 | 212 | 6.07 |
| Glassy / ground glass nuclei | 2832 | 81.12 | 659 | 18.88 |
| Nuclear crowding | 2708 | 77.57 | 783 | 22.43 |
| Wrinkled nuclear membrane | 2650 | 75.91 | 841 | 24.09 |
| Nuclear enlargement | 1923 | 55.08 | 1568 | 44.92 |
| Irregular nuclear contour | 3042 | 87.14 | 449 | 12.86 |
| Nuclear overlapping | 2232 | 63.97 | 1259 | 36.06 |

final classification layer, we replaced the original fully connected layer with a custom output layer consisting of a Layer-Norm [35] followed by a linear layer. This layer outputs the predicted scores for the multi-label classification task, corresponding to the nine binary labels in the dataset. The model was optimized for multi-label classification using sigmoid activations on the final layer and a suitable loss function, allowing it to output probabilities for each label independently.

This architecture is well-suited for the task at hand. Its deep, hierarchical feature extraction capabilities, combined with the fine-tuning process, enable it to capture a cell's features effectively. This makes it a powerful choice for classification tasks due to label imbalance and inter-class correlations.

### 3.7 Loss function optimization

In multi-label classification, especially under severe label imbalance, the design of the loss function plays a critical role in determining both the predictive performance and the reliability of model outputs. Throughout our experiments, we explored and compared several loss functions—Binary Cross Entropy (BCE), Focal Loss, Asymmetric Loss (ASL), and the recently proposed Strictly Proper Asymmetric Loss (SPA) combined with a Label-Pair Regularizer (LPR). Each of these losses addresses different challenges inherent in our dataset, and their comparative analysis provides insight into the trade-offs between accuracy, calibration, and class fairness.

Binary Cross Entropy is the default choice for multi-label learning due to its simplicity and widespread adoption. However, BCE inherently assumes balanced label distributions and equal importance across labels. In our task, where rare morphological traits (e.g., mitosis, nuclear grooves) are severely underrepresented, this assumption leads to biased learning and insufficient attention to rare categories.

Focal Loss mitigates this by down-weighting well-classified examples, thus encouraging the model to focus on hard, often minority-class, examples. However, in small-scale biomedical datasets, its aggressive scaling can amplify label noise and introduce instability during optimization.

Asymmetric Loss (ASL) improves upon Focal Loss by decoupling the focusing factors for positive and negative samples and introducing asymmetric margins. Nonetheless, it lacks an explicit mechanism to counteract overconfident false positives—a problematic behavior in clinical contexts where false alarms on rare traits can lead to unnecessary downstream interventions.

To address this issue of ASL, we adopt the Strictly Proper Asymmetric Loss (SPA), which explicitly penalizes overconfidence in negative predictions. Its formulation introduces a normalization over the predicted probability $p_c$, reducing the gradient when the model exhibits unjustified high confidence:

$$\mathcal{L}_{\text{SPA}} = -\sum_{c=1}^{L} \left[ y_c \log(p_c) + (1 - y_c) \cdot \frac{\log(1 - p_c)}{p_c + \epsilon} \right] \tag{1}$$

As shown in Eq. (1), the loss function directly calibrated predictions. To further enhance semantic consistency among co-occurring morphological traits, we incorporate a Label-Pair Regularizer (LPR). Let $P \subset \{(i, j)\}$ be the set of label pairs with high pointwise mutual information (PMI) observed in the training data. For example, traits such as nuclear hyperchromasia and grooves often appear together. The LPR penalizes inconsistent predictions for such correlated label pairs, as shown in Eq. (2):

$$\mathcal{L}_{\text{LPR}} = \sum_{(i,j) \in P} w_{ij}(p_i - p_j)^2 \tag{2}$$

where $w_{ij}$ denotes the PMI-based correlation strength between labels $i$ and $j$. The final total loss [33] is defined in Eq. (3):

$$\mathcal{L}_{\text{total}} = \mathcal{L}_{\text{SPA}} + \lambda \times \mathcal{L}_{\text{LPR}} \tag{3}$$

This joint formulation improves model performance in two complementary directions: better calibration through SPA and stronger semantic consistency through LPR. SPA effectively reins in overconfident predictions induced by data rebalancing strategies, ensuring calibrated outputs. When coupled with LPR to maintain consistency across correlated traits, the SPA+LPR (with $\lambda = 0.1$) combination emerges as a principled and reliable loss function for imbalanced multi-label classification in biomedical imaging.

### 3.8 Threshold tuning and evaluation

Multi-label classification requires independent binary decisions for each class, making the common threshold of 0.5 suboptimal, especially under class imbalance. In such settings, fixed thresholds can lead to frequent labels' overprediction and rare ones' underprediction. We adopt a per-class threshold tuning strategy using the validation set to mitigate this.

For each class $c$, we determine an optimal threshold $\tau_c^*$ by performing a grid search over $\tau \in [0.1, 0.9]$, selecting the value that maximizes the F1-score as defined in Eq (4):

$$\tau_c^* = \arg \max_{\tau \in [0.1, 0.9]} \text{F1-score}(y_{:c}, \hat{p}_{:c} > \tau) \tag{4}$$

where $\hat{p}_{:c}$ denotes the predicted probability scores and $y_{:c}$ the ground truth labels for class $c$.

This data-driven threshold calibration accounts for the prevalence of each label in the validation set and complements augmentation and sampling strategies that may distort class priors. Adapting thresholds to class-specific characteristics enhances precision and recall across rare and frequent labels. At inference time, final predictions are obtained by binarizing the output probabilities with the corresponding $\tau_c^*$. This step is particularly crucial when using loss functions like SPA, which emphasize probabilistic calibration, and regularizers like LPR, which encourage consistency among correlated labels.

Model performance is evaluated using a comprehensive set of metrics. Alongside micro-averaged F1-score and Exact Match Accuracy (EMA), we report the mean Average Precision (maP) computed both per example and per class (see Eq (5)):

$$\text{maP}_x = \frac{1}{N} \sum_{i=1}^{N} \text{AP}(y_i, \hat{p}_i), \quad \text{maP}_y = \frac{1}{C} \sum_{c=1}^{C} \text{AP}(y_{:c}, \hat{p}_{:c}) \tag{5}$$

where $N$ is the number of instances, $C$ the number of classes, and AP denotes average precision.

This evaluation setup ensures a balanced and reliable assessment of model performance under multi-label imbalance. Per-class threshold tuning, in particular, is a crucial step to align model outputs with real-world decision boundaries, ultimately improving both predictive accuracy and robustness.

## 4 Results

This section comprehensively evaluates the proposed multi-label classification pipeline for single-cell image analysis. The objective is to quantify and analyze the impact of each key component systematically—including the backbone architecture, data augmentation strategies, label imbalance handling, output calibration, and model interpretability.

The analysis is organized into five subsections. First, we benchmark several widely adopted backbone architectures to identify the most suitable one for this domain-specific, imbalanced multi-label task. Second, we conduct an ablation study on different training configurations of ConvNeXt, the backbone selected from the previous phase. The third subsection investigates the effect of loss functions on prediction calibration. The final two subsections examine model interpretability via Grad-CAM and explore the embedding space structure using t-SNE/UMAP to understand the learned representations and decision boundaries better.

## 4.1 Benchmark evaluation of backbone architectures

**4.1.1 Model evaluation in basic pipeline.** We implement seven popular deep learning models to assess the baseline performance of various backbone architectures, as in Table 2. To ensure a fair comparison, all models are trained under the same basic setup: using standard BCE loss, no advanced augmentation (only resizing and normalization), no CutMix, no sample reweighting, no threshold tuning, and fixed training duration (10 epochs).

Table 2 presents the performance of seven backbone architectures on the single-cell multi-label classification task. Swin Transformer leads across all metrics except $maP_x$, with the highest F1 Macro (0.639), F1 Micro (0.717), F1 Weighted (0.712), Exact Match (0.200), and $maP_y$ (0.699), indicating superior handling of rare labels, overall generalization, and label-wise precision across basic settings. DeiT achieves the highest $maP_x$ (0.847), reflecting substantial precision per example. ResNet50 ties with Swin Transformer for Exact Match (0.200) but lags in other metrics (e.g., F1 Micro 0.709, $maP_y$ 0.686). ConvNeXt performs well in F1 Micro (0.712) and F1 Weighted (0.707) but falls behind in F1 Macro (0.627) and Exact Match (0.181). ViT, DeiT, and ViT-B/16 show moderate results, with ViT-B/16 ($maP_y$ 0.659) and ViT ($maP_y$ 0.660) underperforming in precision. EfficientNet-B3 is the weakest, with the lowest scores across all metrics (e.g., F1 Macro 0.565, $maP_y$ 0.613), highlighting its inadequacy for this task.

In general, ResNet50, ConvNeXt-Base, and Swin Transformer are the top three architectures with the most competitive performance across metrics. Swin Transformer consistently leads in most categories, including F1 Macro (0.639), F1 Micro (0.717), F1 Weighted (0.712), Exact Match (0.200), and $maP_y$ (0.699).

**4.1.2 Model evaluation in hybrid pipeline.** We thoroughly evaluated three backbone architectures, ConvNeXt, ResNet50, and Swin Transformer, to identify the most suitable model for our proposed framework, aligning well with our limited dataset. This evaluation was performed within our hybrid learning pipeline, incorporating data augmentation techniques (notably CutMix applied specifically to rare labels), a variant loss function (SPA loss combined with Labelwise Pairwise Regularization, LPR), class-balanced sampling, and threshold tuning. All three architectures were trained and tested under identical conditions on the same dataset, with results reported both per class and globally using standard multi-label evaluation metrics.

Tables 3 and 4 present the detailed per-class classification results and overall performance comparison of the three backbone architectures evaluated on the test set. Table 3, ConvNeXt, ResNet50, and Swin Transformer demonstrate varied strengths across individual labels regarding precision, recall, and F1-score. The overall performance metrics in Table 4 further summarize these results by aggregating classification quality across all classes. These results show that ConvNeXt consistently outperforms both ResNet50 and Swin Transformer regarding overall classification performance. ConvNeXt achieves the highest Micro-F1 score (0.722) and exact match (0.192), both critical metrics for evaluating multi-label prediction reliability. Additionally, it attains the highest mean average precision per example ($maP_x$ = 0.851), indicating its strong ability to rank relevant labels at the sample level correctly. Although Swin Transformer slightly exceeds ConvNeXt in class-level mean average precision ($maP_y$ = 0.708), it shows less stable performance on minority labels such as classes 3, 5, and 7, where F1-scores drop noticeably. Conversely, ConvNeXt maintains more consistent performance across both frequent and rare classes. Furthermore, ConvNeXt demonstrates the best balance between precision and

**Table 2. Performance comparison of backbone architectures on the single-cell multi-label classification task.**

| Backbone | $F1_{macro}$ | $F1_{micro}$ | $F1_{weighted}$ | Exact Match | $maP_x$ | $maP_y$ |
|---|---|---|---|---|---|---|
| ConvNeXt [9] | 0.627 | 0.712 | 0.707 | 0.181 | 0.840 | 0.690 |
| ResNet50 [24] | 0.633 | 0.709 | 0.702 | 0.200 | 0.841 | 0.686 |
| Swin Transformer [38] | **0.639** | **0.717** | **0.712** | **0.200** | 0.839 | **0.699** |
| EfficientNet-B3 [36] | 0.565 | 0.691 | 0.675 | 0.139 | 0.820 | 0.613 |
| ViT [25] | 0.613 | 0.695 | 0.689 | 0.179 | 0.840 | 0.660 |
| DeiT [37] | 0.618 | 0.705 | 0.697 | 0.189 | **0.847** | 0.678 |
| ViT-B/16 [25] | 0.617 | 0.709 | 0.701 | 0.189 | 0.839 | 0.659 |

**Table 3**. Per-class classification report for each backbone on the test set.

| ConvNeXt | | | | ResNet50 | | | Swin Transformer | | |
|---|---|---|---|---|---|---|---|---|---|
| Label | Precision | Recall | F1 | Precision | Recall | F1 | Precision | Recall | F1 |
| 0 | 0.73 | 0.97 | 0.83 | 0.75 | 0.90 | 0.82 | 0.76 | 0.91 | 0.83 |
| 1 | 0.69 | 0.95 | 0.80 | 0.73 | 0.86 | 0.79 | 0.69 | 0.92 | 0.79 |
| 2 | 0.54 | 0.67 | 0.60 | 0.72 | 0.43 | 0.54 | 0.80 | 0.53 | 0.64 |
| 3 | 0.67 | 0.54 | 0.60 | 0.76 | 0.43 | 0.55 | 0.54 | 0.70 | 0.61 |
| 4 | 0.41 | 0.55 | 0.47 | 0.42 | 0.65 | 0.51 | 0.41 | 0.54 | 0.47 |
| 5 | 0.59 | 0.61 | 0.60 | 0.62 | 0.60 | 0.61 | 0.72 | 0.44 | 0.55 |
| 6 | 0.77 | 0.80 | 0.79 | 0.79 | 0.82 | 0.81 | 0.72 | 0.86 | 0.79 |
| 7 | 0.57 | 0.51 | 0.53 | 0.48 | 0.69 | 0.57 | 0.46 | 0.73 | 0.56 |
| 8 | 0.57 | 0.80 | 0.67 | 0.55 | 0.82 | 0.66 | 0.65 | 0.70 | 0.67 |

**Note**: Precision, Recall, and F1-score are reported per class for each model.

**Table 4**. Overall performance comparison across hybrid pipelines.

| Model | $F1_{macro}$ | $F1_{micro}$ | $F1_{weighted}$ | Exact Match | $maP_x$ | $maP_y$ |
|---|---|---|---|---|---|---|
| ConvNeXt | **0.656** | **0.723** | **0.722** | **0.192** | **0.851** | 0.700 |
| ResNet50 | 0.650 | 0.716 | 0.690 | 0.175 | 0.840 | 0.689 |
| Swin Transformer | 0.650 | 0.716 | 0.690 | 0.185 | 0.841 | **0.708** |

recall on a per-sample basis, reflected by its superior sample-wise F1-score. This robustness is particularly advantageous given the noise and class imbalance inherent to single-cell biomedical datasets.

ConvNeXt outperforms ResNet50 and Swin Transformer, particularly in handling imbalanced data. While Swin shows competitive results in basic settings, it struggles when advanced techniques are applied, likely due to its need for large and balanced datasets to utilize hierarchical attention fully. In contrast, ConvNeXt's convolutional bias makes it more stable and better suited for our limited and noisy biomedical dataset.

## 4.2 Ablation study on ConvNeXt configurations

In this section, we conduct an ablation study to systematically evaluate the contribution of each component in our pipeline using ConvNeXt as the backbone. The examined components include data augmentation, CutMix applied to rare labels, weighted sampling, different loss functions (BCE, Focal, ASL, SPA, and SPA with Pairwise Regularizer), and threshold tuning for label-wise classification decision calibration.

Fig 3 presents the results of our ablation study, in which we fix the ConvNeX architecture and vary preprocessing pipelines and loss functions. Each configuration is trained for 10 epochs and evaluated on the test set using standard multi-label classification metrics: macro/micro/weighted F1, exact match accuracy, sample-level mean average precision ($maP_x$), and label-level mean average precision ($maP_y$).

As shown in Fig 3, data augmentation consistently improves performance, particularly for BCE and SPA losses. Combining SPA+PR loss with both augmentation and CutMix yields the best results overall. CutMix is beneficial in selective settings (e.g., BCE with basic transform) but can occasionally reduce performance, emphasizing the need for careful integration based on the loss function. ASL performs poorly across all metrics, with zero exact match accuracy, suggesting it is unsuitable for our small, imbalanced dataset. The best configuration (SPA+PR + augmentation + CutMix) achieves macro F1 = 0.6564, micro F1 = 0.7230, and exact Match = 0.1813, outperforming BCE and Focal loss baselines.

Fig 4 shows the learning curves of this top-performing configuration (SPA+PR with augmentation and CutMix) over 10 epochs. Training loss steadily decreases from 0.61 to 0.419, indicating effective learning, while validation loss remains stable, suggesting no overfitting. F1 scores improve consistently, with F1 Micro reaching 0.720 and F1 Macro 0.647, demonstrating robust performance across labels. Precision (0.744) and recall (0.835) are balanced. AUC rises to 0.911,

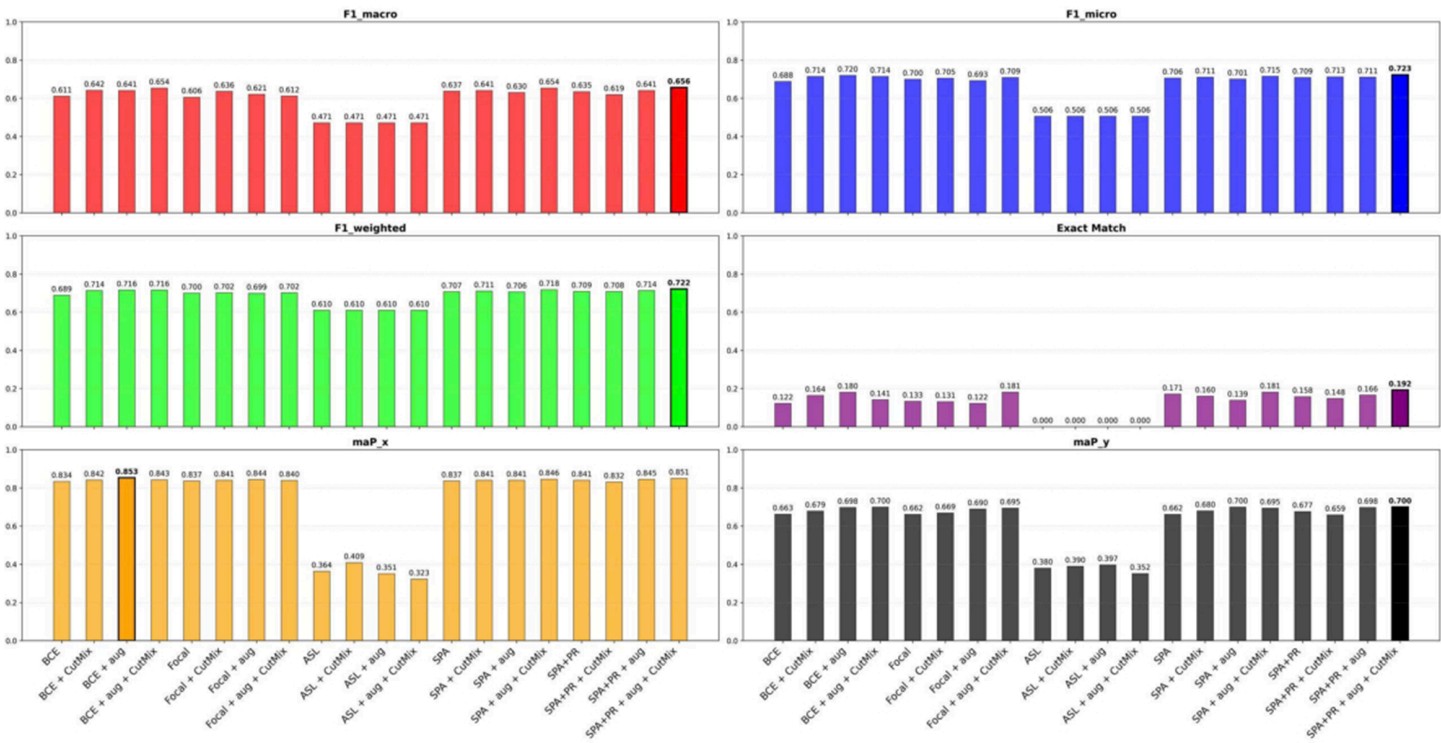

**Fig 3. Performance comparison of various ConvNeXt pipeline configurations.**

and exact match accuracy improves to 0.170. The label-wise mean average precision ($maP_y = 0.829$) slightly exceeds the sample-wise precision ($maP_x = 0.661$), confirming strong label-level discrimination. These results highlight the model's stability and effectiveness for imbalanced multi-label classification.

### 4.3 Calibration analysis

**4.3.1 Calibration performance across loss functions.** To compare the impact of different loss functions (BCE, Focal, ASL, SPA, SPA + PR) on the prediction accuracy and probability calibration of ConvNeXt, we utilized reliability diagrams as presented in Fig 5. This diagram visually represents the relationship between the mean predicted probability (x-axis) and the actual accuracy (y-axis) for different groups of samples. A well-calibrated model will have a curve closely following the dashed diagonal line, indicating a strong agreement between predicted confidence and the true correctness rate.

Fig 5 shows that the curve corresponding to the Focal loss function lies above the ideal calibration line, indicating over-confidence in the predicted probabilities. In contrast, the ASL loss function curve shows under-confidence, with low accuracy, highlighting poor performance and unreliable predictions.

Notably, the BCE, SPA, and SPA+PR curves exhibit similar trends, closely following each other with slight deviations from the ideal calibration line, reflecting moderate calibration performance. SPA+PR aligns most closely with the ideal line, demonstrating the best calibration and providing more reliable probability estimates, making it highly suitable for real-world applications.

**4.3.2 Per-label calibration analysis.** This section explicitly examines the calibration performance of the SPA+PR loss function label. By generating reliability diagrams for each of the nine individual labels, we aim to identify potential

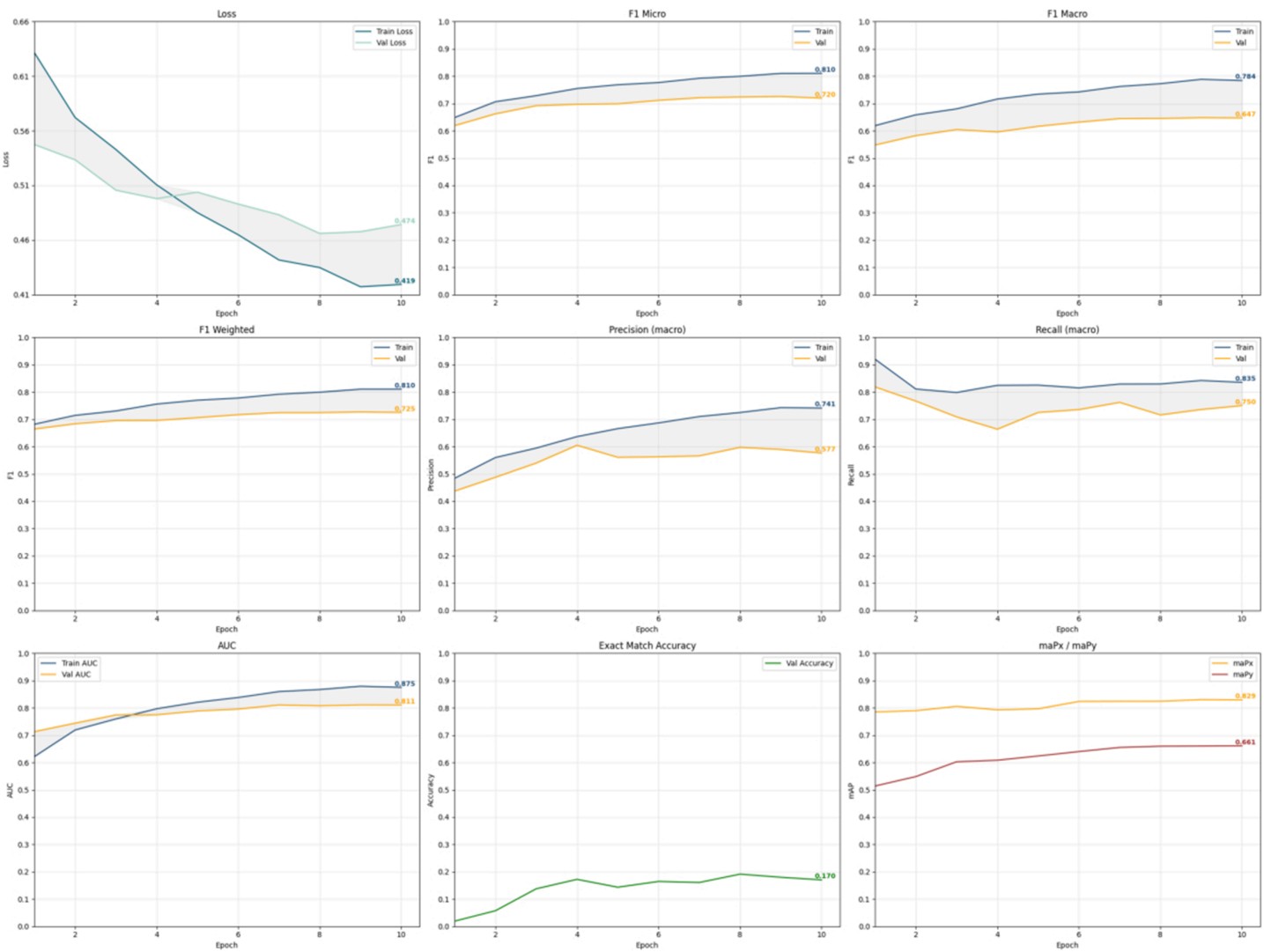

**Fig 4**. Learning curves for top-performing model.

variations in calibration across different classes that an overall calibration assessment might mask. The following figures illustrate the calibration curves for each label.

The calibration performance for each label can be summarized as follows:

- **Class 0:** Over-confident at low/medium probabilities, better calibrated at high probabilities (~0.7).
- **Class 1:** Over-confident at low/medium probabilities (<0.6), improves at high probabilities (>0.6).
- **Class 2:** over-confident across the entire probability range.
- **Class 3:** Over-confident, especially at medium probabilities (~0.5-0.6).
- **Class 4:** Over-confident at medium probabilities (~0.5-0.7), slightly under-confident at low probabilities.
- **Class 5:** Over-confident at medium probabilities (~0.5-0.7), under-confident at high probabilities (~0.8).
- **Class 6:** Unstable calibration shows over-confidence and under-confidence at different probability ranges.
- **Class 7:** over-confident at medium probabilities (~0.3-0.4), tends to improve at high probabilities (sparse data).

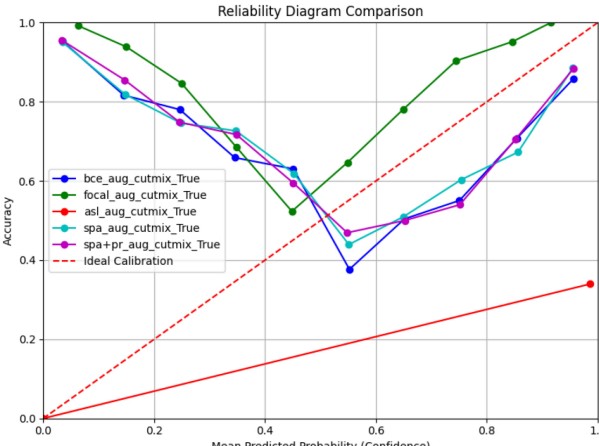

**Fig 5**. **Reliability diagram comparison of different configurations.**

- **Class 8:** Under-confident at low/medium probabilities (<0.6), improves and potentially over-confident at high probabilities (>0.6).

According to the reliability diagram, SPA+PR achieved better overall calibration; however, this improvement does not uniformly extend to all labels. As detailed in Fig 6, some individual labels persistently exhibit over-confidence, most notably for predictions in the medium probability range, challenging the perception of universally improved reliability. This discrepancy is likely attributed to the imbalance of the multi-label classification task despite our effort to augment data for rare labels. Dominant labels may exert a greater influence during optimization, leading to suboptimal calibration for less frequent or more challenging labels. Nevertheless, the overall calibration achieved by SPA+PR remains superior to other tested loss functions. This suggests that SPA+PR offers a better balance, although further research into techniques like post-hoc calibration may be necessary to enhance calibration at the individual label level.

### 4.4 Confusion matrices for individual labels

The confusion matrices (Fig 7) provide a detailed assessment of the model's performance across all individual labels. Certain labels, such as *Intranuclear inclusion* (label 2) and *Irregular nuclear contour* (label 7), show strong performance with high true positive (TP) and true negative (TN) counts, indicating the model can effectively distinguish both the presence and absence of these features. In contrast, labels like *Nuclear elongation* (label 0) and *Nuclear grooves* (label 1) exhibit a high number of false positives (FP), suggesting the model tends to over-predict these features. While it often correctly identifies their absence (high TN), it struggles to confirm their presence (lower TP) for the remaining labels—including *Glassy/ground glass nuclei* (label 3), *Nuclear crowding* (label 4), *Wrinkled nuclear membrane* (label 5), *Nuclear enlargement* (label 6), and *Nuclear overlapping* (label 8)—the confusion matrices indicate moderate performance. These labels show a mixed pattern of false positive (FP) and false negatives (FN), reflecting the model's limited ability to detect these features confidently.

The high false positive rates partly arise from subtle visual characteristics and overlapping morphologies, which inherently create ambiguity in classification. In addition, the imbalance between false positives and false negatives across several labels suggests that the model adopts a conservative decision boundary. Beyond these intrinsic challenges, other factors also contribute to misclassifications. First, the dataset size (3,419 single-cell images) is relatively small compared to typical deep learning benchmarks. Although no severe overfitting was observed, the gap between training and validation metrics suggests limited generalizability, particularly for rare nuclear patterns. Second, the strong class imbalance

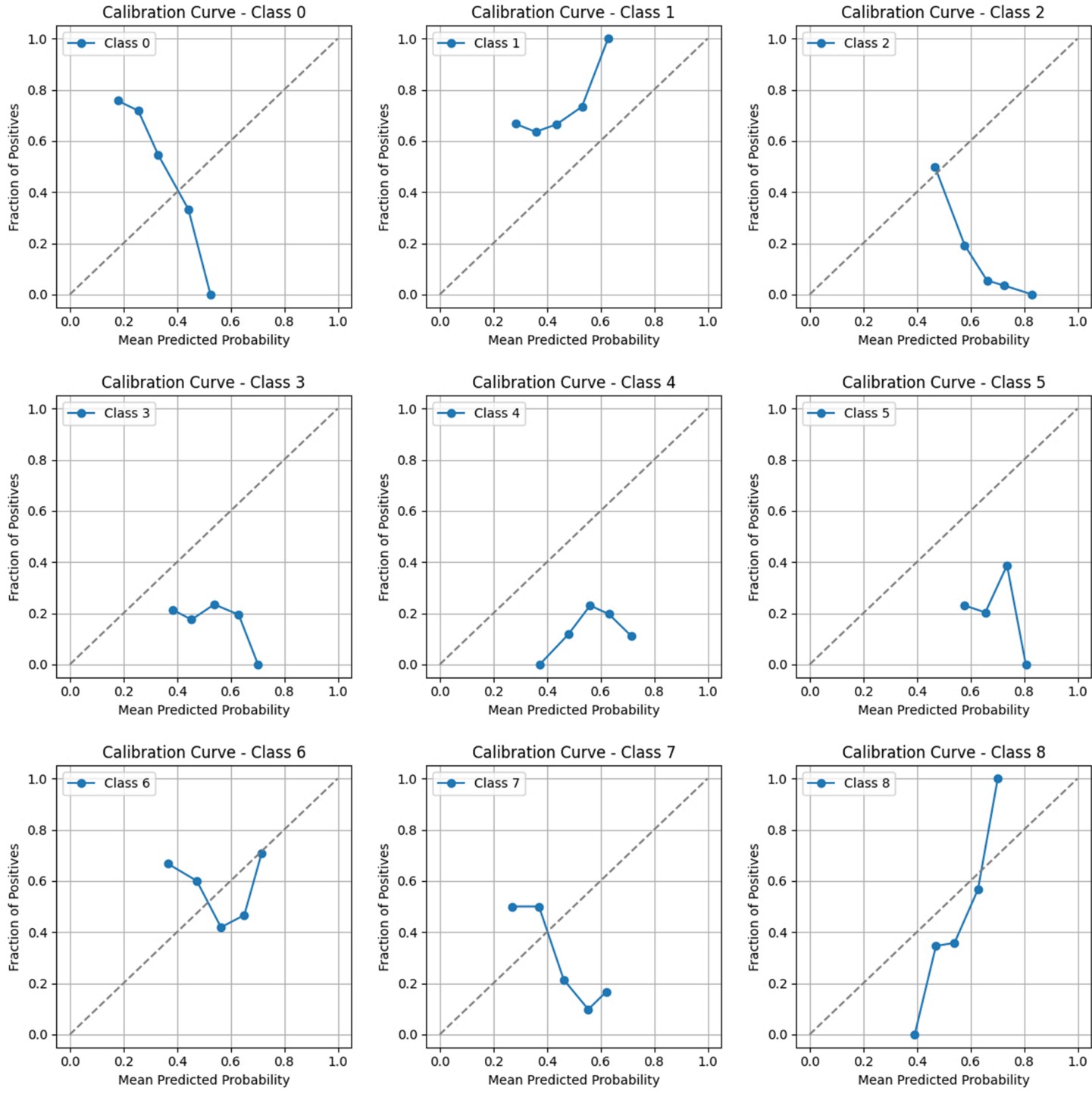

**Fig 6**. Label-wise calibration curves for each of the nine classes.

in the dataset (see Table 1) biases the model toward more frequent features, leading to high false positive rates for common labels and reduced sensitivity for rare ones—for example, intranuclear inclusions have only 212 positive samples

PLOS Digital Health

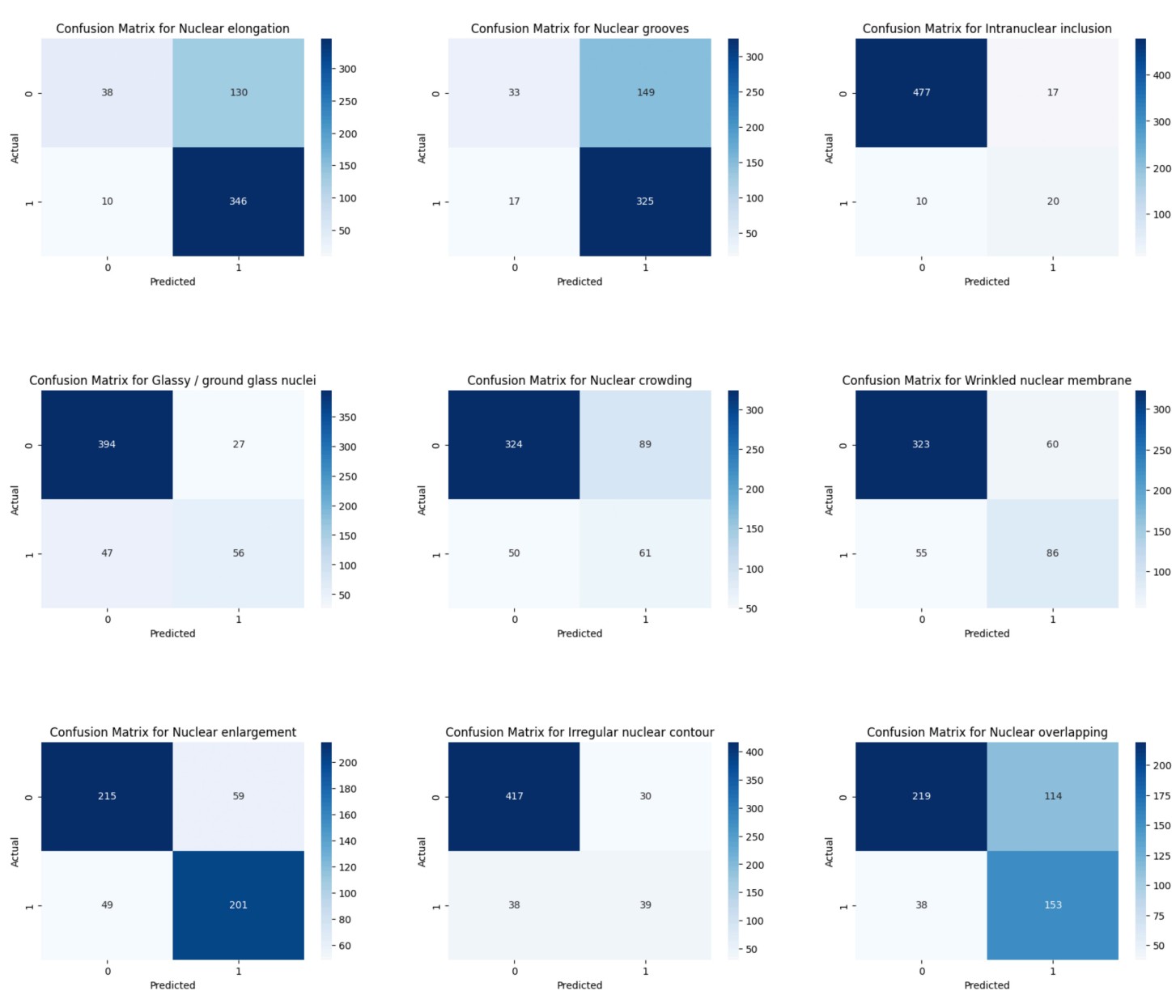

**Fig 7**. Confusion matrices for each label.

compared with 2,445 for nuclear elongation. Third, certain features share overlapping morphology, such as grooves and irregular contours, which even ConvNeXt's feature extraction struggles to disentangle. From a cytological perspective, both features involve irregularities along the nuclear membrane: grooves appear as shallow linear indentations, while irregular contours manifest as uneven or distorted nuclear outlines. When visualized at the single-cell level, these patterns can appear strikingly similar, particularly in borderline cases. This challenge is further compounded by the relatively small input resolution of our dataset (cropped single-cell images of approximately 80×80 pixels), which may obscure subtle boundaries and fine membrane details. While the ConvNeXt architecture effectively captures global texture and shape, it may not fully resolve these fine-grained structures, making it challenging to reliably separate grooves from irregular contours.

Taken together, these limitations indicate that the observed false positives and false negatives are not solely due to inherent visual ambiguity but also reflect dataset constraints and architectural biases. This underscores the need for future work on expanding and balancing the dataset, designing targeted augmentation for rare morphologies, and exploring models that better leverage spatial context.

### 4.5 ConvNeXt interpretation via grad-CAM

Understanding the decision-making process of a model is essential in high-stakes biomedical applications. To gain interpretability in our ConvNeXt model's behavior, we applied Grad-CAM (Gradient-weighted Class Activation Mapping), which visualizes the spatial regions most influential to the model's output for each label. This allows us to assess whether the model focuses on histopathologically relevant features crucial for clinical trust and biological plausibility. We present qualitative analysis on two representative test samples with exact-match predictions, illustrating a strong alignment between the model's attention and key nuclear characteristics.

**Case 1:** `[1 1 0 0 0 0 0 0 0]` This sample, detailed in Table 5, includes two positive labels—*Nuclear elongation* and *Nuclear grooves*—and seven negatives. The model correctly predicted all labels. This relatively sparse prediction allows us to closely examine the focused activation regions for each detected class with minimal label overlap. The corresponding Grad-CAM visualizations are provided in Fig 8, which highlight strong attention over the elongated axis of the nucleus (for *Nuclear elongation*) and over subtle indentations of the nuclear membrane (for *Nuclear grooves*). These regions align well with morphological cues typically used by human experts.

**Case 2:** `[0 1 0 1 0 1 1 1 1]` Fig 9 presents a detailed visualization of the model's attention when simultaneously predicting multiple morphological features in a single cell image. This complex case offers valuable insight into how the model handles overlapping characteristics. The following six labels were correctly predicted, and we analyze each along with the corresponding Grad-CAM heatmaps:

- **Nuclear grooves (p = 0.46):**
  The heatmap highlights shallow indentations on the nuclear membrane, aligning with the expected appearance of nuclear grooves—subtle invaginations or furrows on the nuclear surface. However, the relatively low confidence ($p = 0.46$) suggests some ambiguity, possibly due to visual subtlety or overlapping features.
- **Glassy / ground glass nuclei (p = 0.87):**
  Activation is spread across the interior of the nucleus, indicating sensitivity to diffuse textural changes consistent with chromatin alterations that produce a glassy or "ground glass" appearance. This diffuse pattern aligns with the morphological definition of the feature.

**Table 5**. textbfGrad-CAM: Groundtruth and model's prediction yielding exact match result.

| Index | Label Name | Ground Truth | Prediction |
|---|---|---|---|
| 0 | Nuclear elongation | 1 | 1 |
| 1 | Nuclear grooves | 1 | 1 |
| 2 | Intranuclear inclusion | 0 | 0 |
| 3 | Glassy / ground glass nuclei | 0 | 0 |
| 4 | Nuclear crowding | 0 | 0 |
| 5 | Wrinkled nuclear membrane | 0 | 0 |
| 6 | Nuclear enlargement | 0 | 0 |
| 7 | Irregular nuclear contour | 0 | 0 |
| 8 | Nuclear overlapping | 0 | 0 |
| **Match (Overall)** | **True** | | |

## (a) Sample 1

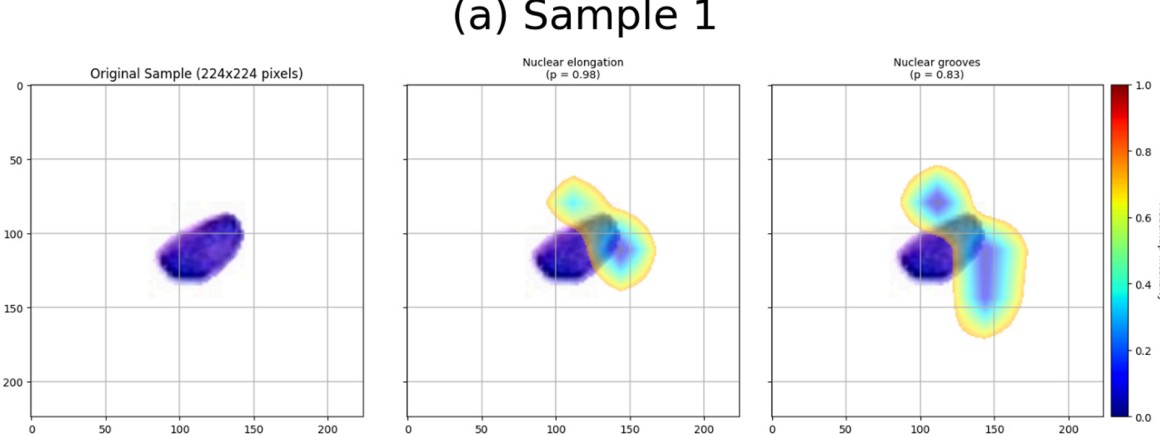

## (b) Sample 2

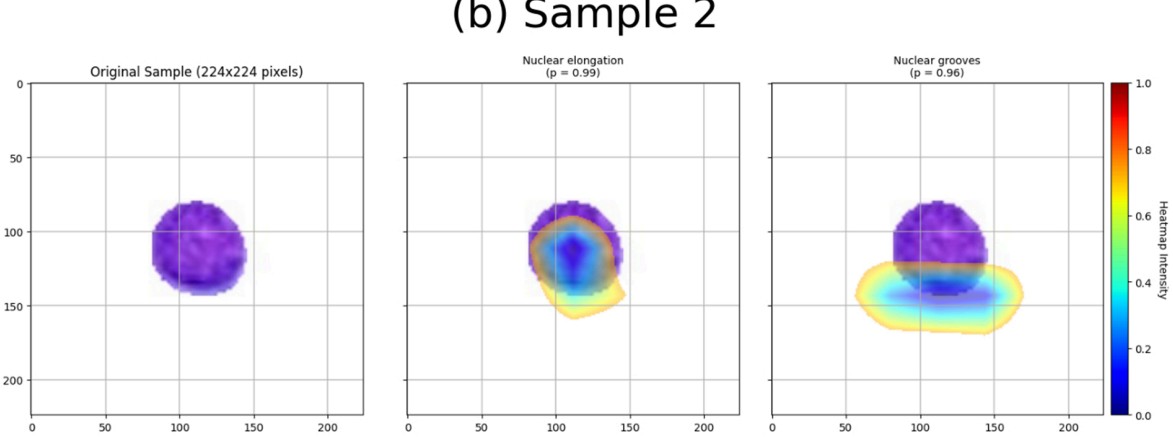

## (c) Sample 3

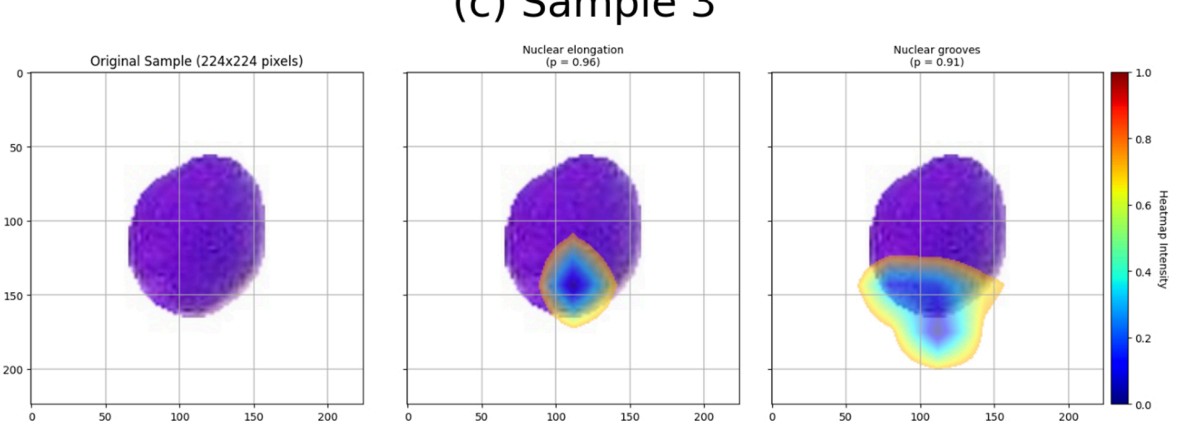

**Fig 8**. GradCam visualization for label [1 1 0 0 0 0 0 0 0 0] with exact match prediction.

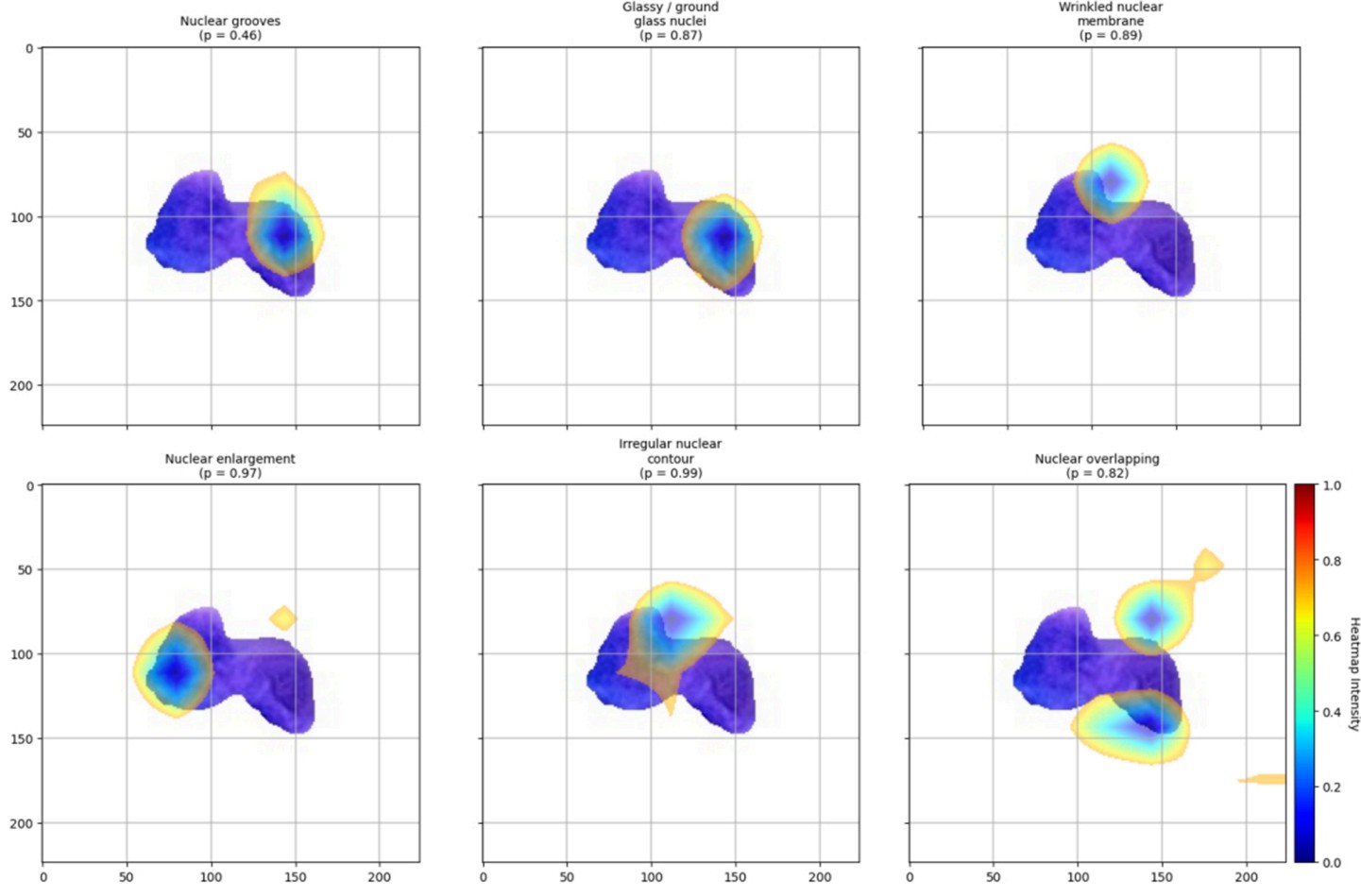

**Fig 9**. **GradCam visualization for the label [0 1 0 1 0 1 1 1 1] with exact match prediction.**

- **Wrinkled nuclear membrane (p = 0.89):**
    The model focuses attention along the periphery of the nucleus, revealing activation corresponding to fine undulations or folds in the nuclear membrane. The high confidence reflects the model's ability to detect edge-based morphological variation.
- **Nuclear enlargement (p = 0.97):**
    Heatmap activation spans the entire nuclear region, consistent with the label's semantic meaning. This suggests the model assesses the nucleus's overall size and scale, and the very high confidence implies substantial certainty in the presence of this feature.
- **Irregular nuclear contour (p = 0.99):**
    The attention map, including the boundary and inner nuclear areas, indicates that the model considers shape irregularities and internal structural deformations. The near-certain prediction shows that this feature is visually prominent in the image.
- **Nuclear overlapping (p = 0.82):**
    The highlighted regions correspond to where nuclei cluster or overlap, validating the model's focus on nuclear proximity and crowding. This spatial pattern matches the label's description and suggests learned sensitivity to cell density.

These visualizations demonstrate the model's ability to localize and distinguish multiple morphological features within a single cell. The heatmaps' varying intensity and spatial distribution suggest that the model selectively attends to relevant nuclear regions depending on the feature. Differences in prediction confidence further reflect each label's inherent visual clarity or ambiguity. Overall, the Grad-CAM outputs provide intuitive explanations for the model's decisions, highlighting its reliance on interpretable cues such as shape, texture, and nuclear boundaries.

Furthermore, Grad-CAM adds an important layer of transparency by showing whether the attention of the model aligns with regions that are meaningful for cytological assessment. This is especially critical in medical applications, where explainability directly impacts clinical trust. Notably, the attention maps consistently focus on the nuclei themselves rather than the white background added during preprocessing. This observation indirectly validates our data preparation strategy of cropping individual cells and embedding them in a neutral background, since it shows that the model does not rely on artificial regions outside the cell. When the highlighted areas correspond to features that pathologists typically examine, the visualization reassures that the model is learning relevant biological signals rather than random patterns. Such interpretability also allows clinicians to review and cross-check the model's reasoning, helping to reduce the black-box perception of deep learning, which refers to the concern that models generate predictions without offering understandable reasons. Although Grad-CAM remains qualitative and does not provide strict quantitative validation, it serves as a valuable bridge between computational results and clinical judgment, supporting the safer translation of AI systems into practice.

## 5 Discussion

This study introduces a novel, expert-annotated single-cell image dataset for thyroid cancer diagnosis and establishes robust deep learning baselines for multi-label classification of nuclear features. Our analysis provides critical insights into the efficacy of the proposed methodologies and highlights the inherent challenges of working with new cytological data.

### 5.1 Key findings

A primary finding of this research is that, among the multiple backbones evaluated, the ConvNeXt architecture consistently produced strong and stable results when integrated with advanced data-handling techniques. While attention-based models such as the Swin Transformer performed competitively in basic setups, ConvNeXt demonstrated greater stability and robustness when sophisticated strategies were applied to address the severe class imbalance characteristic of biomedical datasets. This suggests that the inductive biases of modern convolutional architectures may be particularly advantageous for limited and noisy datasets, whereas attention-based models often require larger and more balanced data to perform optimally.

Another significant contribution is the demonstrated effectiveness of the Strictly Proper Asymmetric Loss (SPA) combined with a Label-Pair Regularizer (LPR). Our ablation study revealed that this loss function, when coupled with data augmentation and CutMix, yielded the best overall classification performance, achieving a micro F1 score of 0.723. Calibration analysis further showed that SPA+PR improved the reliability of the model's probability estimates compared to standard losses such as BCE and Focal Loss. In a clinical context, where trustworthiness is paramount, this enhanced calibration represents a critical advantage.

Finally, our interpretability analysis using Grad-CAM provides compelling evidence that the model learns biologically relevant features. The activation maps consistently highlight nuclear regions that align with the morphological characteristics examined by pathologists, such as nuclear contour, texture, and shape. This increases confidence in the model's predictions and validates our data preparation strategy, showing that the model successfully learned to focus on cellular features while ignoring artificial background.

## 5.2 Limitations

Despite these promising results, several limitations must be acknowledged. The first concerns the dataset itself: although meticulously annotated by experts, it remains relatively small (3,419 images) and originates from a single medical institution. This narrow scope may limit the generalizability of our models to data from other hospitals with different staining protocols or imaging devices, thereby introducing potential domain shift.

Another limitation lies in the severe class imbalance that persists despite multiple mitigation strategies. This imbalance is reflected in varied performance across labels, with lower sensitivity for rare but clinically significant features. Per-label calibration analysis further revealed that improvements in overall calibration via SPA+PR did not uniformly extend to all labels, likely due to the limited representation of certain classes.

A third challenge relates to misclassifications caused by both inherent visual ambiguity and the relatively low image resolution. Features such as nuclear grooves and irregular contours often share overlapping morphological characteristics, and cropped single-cell images at approximately 80×80 pixels may obscure subtle membrane details needed for confident classification.

Finally, while Grad-CAM offered valuable visual insights, it remains a qualitative interpretation tool. Its outputs should be understood as plausibility assessments rather than definitive explanations, as the method does not provide strict quantitative validation of feature attribution.

Taken together, these limitations underscore opportunities for future work to enhance model reliability, address rare-label challenges, and better capture subtle morphological features.

## 6 Conclusion and future work

In this study, we built an expert-annotated single-cell image dataset for thyroid cancer diagnosis, comprising 3,419 images labeled with nine clinically relevant nuclear features. We established robust deep learning baselines for multi-label classification, leveraging state-of-the-art architectures such as ConvNeXt, Vision Transformers, and ResNet. To address class imbalance, we employed techniques including conditional CutMix, weighted sampling, and the SPA loss with Label Pairwise Regularization. Our experiments demonstrate that the ConvNeXt architecture, optimized with SPA+PR loss, data augmentation, and CutMix, achieves a micro F1 score of 0.723 on the test set, effectively handling multi-label classification with imbalanced data. Calibration analysis indicates reliable probability estimates, and Grad-CAM visualizations confirm that the model focuses on pathologically relevant regions, enhancing interpretability and clinical applicability.

Looking ahead, our future work will focus on four key areas to enhance the clinical applicability of our pipeline. Firstly, we will systematically evaluate post-hoc calibration methods, such as temperature scaling and isotonic regression, on rare labels to improve the trustworthiness of probability estimates. Secondly, we plan to significantly expand our dataset by collaborating with additional hospitals, integrating new single-cell data, and standardizing imaging metadata, which will enable us to re-evaluate attention-based architectures, such as Swin Transformers. Thirdly, we will implement advanced data augmentation techniques, including morphology-preserving transformations and GAN-based synthetic sample generation for underrepresented features. Finally, we will explore ensemble and fusion strategies, combining predictions from individual backbones using weighted averaging and trainable late fusion to reduce variance and improve robustness. These comprehensive steps are expected to mitigate class imbalance, refine rare-label calibration, and enhance overall model robustness for clinical deployment.

We envision that models like ours could be integrated into telepathology platforms to support remote diagnostics, particularly in under-resourced regions where access to trained specialists is limited. By providing automated assistance in analyzing complex cellular features, such systems could reduce diagnostic delays, alleviate the workload of pathologists, and enhance the consistency of clinical decision-making. Ultimately, this integration has the potential to expand access to high-quality diagnostic expertise while maintaining strict oversight, transparency, and accountability in medical practice

## Author contributions

**Data curation:** Van De Nguyen, Kim Giap Hoang, Huyen Tram Vu.

**Formal analysis:** Quang Huy Nguyen, DO Thanh Ha.

**Investigation:** Quang Huy Nguyen.

**Methodology:** Quang Huy Nguyen.

**Supervision:** DO Thanh Ha.

**Validation:** Quang Huy Nguyen.

**Writing – original draft:** Quang Huy Nguyen, DO Thanh Ha.

**Writing – review & editing:** Quang Huy Nguyen, DO Thanh Ha.

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
