## [Decision Letter · Decision Letter 0]

15 Aug 2025

PDIG-D-25-00315A Novel Expert-Annotated Single-Cell Dataset for Thyroid Cancer Diagnosis with Deep Learning BenchmarksPLOS Digital Health Dear Dr. Thanh Ha, Thank you for submitting your manuscript to PLOS Digital Health. After careful consideration, we feel that it has merit but does not fully meet PLOS Digital Health's publication criteria as it currently stands. Therefore, we invite you to submit a revised version of the manuscript that addresses the points raised during the review process. Please submit your revised manuscript within 30 days Sep 14 2025 11:59PM. If you will need more time than this to complete your revisions, please reply to this message or contact the journal office at digitalhealth@plos.org. Please include the following items when submitting your revised manuscript: * A rebuttal letter that responds to each point raised by the editor and reviewer(s). You should upload this letter as a separate file labeled 'Response to Reviewers'. This file does not need to include responses to any formatting updates and technical items listed in the 'Journal Requirements' section below.* A marked-up copy of your manuscript that highlights changes made to the original version. You should upload this as a separate file labeled 'Revised Manuscript with Track Changes'.* An unmarked version of your revised paper without tracked changes. You should upload this as a separate file labeled 'Manuscript'. If you would like to make changes to your financial disclosure, competing interests statement, or data availability statement, please make these updates within the submission form at the time of resubmission. Guidelines for resubmitting your figure files are available below the reviewer comments at the end of this letter. We look forward to receiving your revised manuscript. Kind regards, Sulaf Assi, PhDAcademic EditorPLOS Digital Health Leo Anthony CeliEditor-in-ChiefPLOS Digital Healthorcid.org/0000-0001-6712-6626 **Journal Requirements:**

1. Please provide a complete Data Availability Statement in the submission form, ensuring you include all necessary access information or a reason for why you are unable to make your data freely accessible. If your research concerns only data provided within your submission, please write "All data are in the manuscript and/or supporting information files" as your Data Availability Statement.

2. We ask that a manuscript source file is provided at Revision. Please upload your manuscript file as a .doc, .docx, .rtf or .tex.

3. Your manuscript is missing the following sections: [insert missing section]. Please ensure these are present, and in the correct order, and that any references to subheadings in your main text are correct. An outline of the required sections can be consulted in our submission guidelines here:

https://journals.plos.org/digitalhealth/s/submission-guidelines#loc-parts-of-a-submission

4. Please provide separate figure files in .tif or .eps format.

**Additional Editor Comments (if provided):** Please revise the manuscript taking into account the reviewers' recommendations**Reviewers' Comments:**

Reviewer #1: This is a strong and well-conducted study that makes a substantial contribution to the field of AI-assisted cytology. The creation and annotation of a novel single-cell dataset, combined with benchmarking of deep learning models using modern architectures and loss functions, addresses important gaps in the literature. The paper is particularly commendable for its focus on interpretability (Grad-CAM), rigorous calibration evaluation (SPA+PR), and rare label augmentation (CutMix). The inclusion of both ablation studies and confusion matrix analysis shows a high degree of scientific thoroughness.

To further strengthen the manuscript, I recommend:

a. Clearly confirm that the dataset (and ideally model code) is publicly accessible via a repository, with documentation.

b. Include inter-observer agreement or annotation consistency metrics if possible, to validate annotation quality.

c. Add a brief discussion on the dataset’s generalizability and its limitations due to the single-institution source.

d. Consider extending the ethical discussion to address future clinical applications and implications.

Recommendation:

ACCEPT - The manuscript is technically rigorous, well-executed, and highly relevant. Only small clarifications are needed to meet PLOS Digital Health’s standards on data availability and annotation validation + data base generalizability. I believe this can be addressed during the publication and finalization process for the manuscript.

Detailed Review:

1.The study addresses an important global challenge with strong relevance to researchers in medical imaging, pathology, and machine learning. It introduces a novel, expert-annotated AI-driven single-cell cytology dataset for thyroid cancer diagnosis. It fills a gap in publicly available, high-quality single-cell image datasets and benchmarks several deep learning models tailored for this task. The integration of SPA+PR loss, CutMix augmentation for rare labels, and detailed calibration analysis adds innovative depth. By focusing on model interpretability and calibration, the work also responds to current concerns about the clinical trustworthiness of AI in health.

2. The paper is well-structured and methodologically detailed. The segmentation and annotation pipeline is clearly described, and ethical approval was obtained. Results are thoroughly analyzed using appropriate metrics: micro/macro F1, exact match, AUC, and calibration plots. Grad-CAM visualizations and confusion matrices support the findings. Benchmark comparisons across model architectures and loss functions are carefully explained. However, while well executed, the study is limited to a single institution and could benefit from a clearer discussion of inter-rater reliability or dataset validation protocols by independent pathologists.

3.The dataset and accompanying benchmarks provide a valuable, reproducible resource for the AI and medical imaging community. The study is technically accessible and well-explained, with clear implications for clinical use and future research. The authors also describe the dataset preparation process and how it can be expanded. The manuscript provides detailed experimental protocols and includes extensive tables, model architecture settings, and evaluation metrics. Though the manuscript references dataset availability, public access to the full dataset and code (e.g., via GitHub or Zenodo) should be clarified to meet full open science standards.

4. The writing is clear, technically accurate, and organized logically. Minor grammatical polishing could improve flow, but overall the manuscript is well-written.

Reviewer #2: Summary: The manuscript presents a novel single-cell cytology dataset for thyroid cancer diagnosis, comprising 3,419 images annotated with nine clinically significant nuclear features. It establishes deep learning baselines using architectures like ConvNeXt, Vision Transformers, and ResNet-50, addressing challenges such as class imbalance and multi-label dependencies through techniques like conditional CutMix, weighted sampling, and Strictly Proper Asymmetric Loss (SPA) with Label Pairwise Regularization (LPR). The study achieves a micro F1 score of 0.723 with ConvNeXt and provides interpretable insights via Grad-CAM visualizations. The dataset preparation, ethical considerations, and comprehensive evaluation make this a valuable contribution to automated cytological analysis.

Recommendations for Minor Revisions:

1. Limited Discussion of Model Limitations: While the study highlights high false positive rates for certain features due to visual ambiguities (section 4.4), it does not sufficiently discuss specific limitations of the ConvNeXt model or potential reasons for misclassifications beyond visual similarity (e.g., dataset size constraints or feature extraction challenges).

o Recommendation: Expand the discussion in Section 4.5 to include a deeper analysis of model limitations, such as the impact of the dataset’s relatively small size (3,419 images) or potential overfitting to common labels. Suggest specific strategies for future improvements (e.g., transfer learning or larger datasets).

2. Future Work Specificity: The future work section mentions improving calibration for rare labels and expanding the dataset but lacks specificity on how these will be achieved.

o Recommendation: Provide more concrete plans for future work, such as specific data augmentation techniques, potential collaborations for dataset expansion, or advanced calibration methods to be explored.

3. The use of Grad-CAM visualizations is a highlight, as it bridges the gap between technical performance and clinical interpretability, which should be emphasized in the revised manuscript.

4. Include a limitation section.

**Figure resubmission:**While revising your submission, please upload your figure files to the Preflight Analysis and Conversion Engine (PACE) digital diagnostic tool, https://pacev2.apexcovantage.com/. PACE helps ensure that figures meet PLOS requirements. To use PACE, you must first register as a user. Registration is free. Then, login and navigate to the UPLOAD tab, where you will find detailed instructions on how to use the tool. If you encounter any issues or have any questions when using PACE, please email PLOS at figures@plos.org. Please note that Supporting Information files do not need this step. If there are other versions of figure files still present in your submission file inventory at resubmission, please replace them with the PACE-processed versions.**Reproducibility:**To enhance the reproducibility of your results, we recommend that authors of applicable studies deposit laboratory protocols in protocols.io, where a protocol can be assigned its own identifier (DOI) such that it can be cited independently in the future. Additionally, PLOS ONE offers an option to publish peer-reviewed clinical study protocols. Read more information on sharing protocols at https://plos.org/protocols?utm_medium=editorial-email&utm_source=authorletters&utm_campaign=protocols

---

## [Editor Report · Decision Letter 1]

17 Nov 2025

A Novel Expert-Annotated Single-Cell Dataset for Thyroid Cancer Diagnosis with Deep Learning Benchmarks

PDIG-D-25-00315R1

Dear PhD Thanh Ha,

We are pleased to inform you that your manuscript 'A Novel Expert-Annotated Single-Cell Dataset for Thyroid Cancer Diagnosis with Deep Learning Benchmarks' has been provisionally accepted for publication in PLOS Digital Health.

Best regards,

Sulaf Assi, PhD

Academic Editor

PLOS Digital Health